# High-Intensity Interval Training upon Cognitive and Psychological Outcomes in Youth: A Systematic Review

**DOI:** 10.3390/ijerph18105344

**Published:** 2021-05-17

**Authors:** Ana R. Alves, Renata Dias, Henrique P. Neiva, Daniel A. Marinho, Mário C. Marques, António C. Sousa, Vânia Loureiro, Nuno Loureiro

**Affiliations:** 1Department of Arts, Humanities and Sports, School of Education, Polytechnic Institute of Beja, 7800-295 Beja, Portugal; vloureiro@ipbeja.pt (V.L.); nloureiro@ipbeja.pt (N.L.); 2Research Center in Sports Sciences, Health Sciences and Human Development, CIDESD, 5001-801 Vila Real, Portugal; henriquepn@gmail.com (H.P.N.); dmarinho@ubi.pt (D.A.M.); mariomarques@mariomarques.com (M.C.M.); antonio_carlossousa@hotmail.com (A.C.S.); 3Laboratory of Physical Activity and Health, School of Education, Polytechnic Institute of Beja, 7800-295 Beja, Portugal; 14743@alunos.ipbeja.pt; 4Department of Sport Sciences, University of Beira Interior, 6201-001 Covilhã, Portugal; 5Research Center in Physical Activity, Values Promotion and Education, HUM-954 Huelva, Spain; 6The Instituto de Saúde Ambiental Research Center (ISAMB), 1649-026 Lisboa, Portugal

**Keywords:** exercise, performance, concentration, attention, well-being, self-concept, evaluation, systematic review

## Abstract

Development of innovative and time-efficient strategies to involve youth in physical activity is pivotal in the actual inactivity pandemic. Moreover, physical activity may improve academic performance, of great interest for educators. This present systematic review aimed to analyze the effects of high-intensity interval training (HIIT) on cognitive performance and psychological outcomes in youth. A database search (Web of Science, PubMed, Scopus, and PsycINFO) for original research articles was performed. A total of eight articles met the inclusion criteria, and the Cochrane risk of bias tool was used. The studies’ results were recalculated to determine effect sizes using Cohen’s d. Different HIIT interventions reported improvements on cognitive performance at executive function (d = 0.75, +78.56%), linguistic reasoning (d = 0.25, +7.66%), concentration (d = 0.71, +61.10%), selective attention (d = 0.81, +60.73%), non-verbal and verbal abilities (d = 0.88, +47.50%; d = 1.58, +22.61%, respectively), abstract reasoning (d = 0.75, +44.50%), spatial and numerical abilities (d = 37.19, +22.85%; d = 1.20, +8.28%, respectively), and verbal reasoning (d = 1.00, +15.71%) in youth. Regarding psychological outcomes, HIIT showed higher self-concept (d = 0.28, +8.71%) and psychological well-being in boys and girls (d = 0.73, +32.43%, d = 0.39, +11.58%, respectively). To sum up, HIIT interventions between 4–16 weeks, for 8–30 min/session, at ≥85% maximal heart rate, would provide positive effects on cognitive performance and psychological outcomes in youth.

## 1. Introduction

Consistent participation in physical activity is associated with a widespread range of physical health benefits for young people, including physiological and psychological benefits related to an active lifestyle [1,2]. Literature suggests that physical activity provides a positive effect on neurocognitive (i.e., attention, concentration) and behavioral (i.e., anxiety, psychological stress, depression) outcomes in youth [3,4]. Moreover, in recent years, a growing number of studies have also reported physical fitness as an influent intermediary of the effects of exercise training on cognition functions and academic performance, through direct and indirect psychological, physiological, and learning methods [5,6]. The conception that greater levels of physical fitness may improve attention, concentration, thinking and consequently academic performance, has a high level of interest for educators and physical educators [7].

Even with the extensive benefits of an active lifestyle, physical inactivity amoung young people is predominant [8,9], and tendencies report a secular weakness in adolescents’ physical fitness levels [10,11]. Assuming that several mechanisms of life change (i.e., increased physical activity, behavioral activation, autonomy, self-efficacy, self-esteem) are frequently supported by concepts associated with enthusiasm or motivation, the employment of existing research and theory may be helpful when drawing novel methodologies [12]. The previous evidence highlights the importance of developing innovative and time-efficient strategies to involve adolescents in physical activity, providing health benefits and effective solutions to this inactivity pandemic.

Accepted as a time-efficient method of achieving the health benefits of physical activity, high-intensity interval training (HIIT) has emerged as a useful and efficacious alternative to the traditional training methods [13,14]. HIIT is comprised of different interval protocols, but generally involves short intervals (≤45 s) of bouts of maximal sprints in high intensity (>85% maximal heart rate) combined with recovery breaks (<60 s) [15,16]. A great curiosity of HIIT is that it represents a method that requires no or minimal equipment, being completed in a short period of time. Moreover, it seems to provide similar physiological adaptations when compared to longer sessions of traditional training methods [15,16,17]. Furthermore, younger populations may consider short bouts of vigorous-intensity exercise more likely, desirable and simple to follow when compared to traditional moderate-intensity exercises [18]. Additionally, involving children and adolescents in activities that could be pleasurable may support the implementation of healthy habits (e.g., remaining physically active), and the development of self-reliant physical activity to be sustained into adulthood [19].

An emergent body of literature supports the feasibility and efficacy of HIIT on improving cognitive function and psychological variables (i.e., depression, emotional wellbeing, sleep quality) in the young population [20,21,22,23,24].

Cognitive performance could be described by several variables concerning executive functions such as concentration, selective attention or working memory [25]. Psychological outcomes are obtained by variables describing behavioral actions such as anxiety, depression, distress, well-being and self-efficacy [26,27]. Low level of cognitive performance during youth has been associated with psychological concerns such as unkind emotions. Those emotions could produce depressive feelings, unhappiness or harmful interpretations of the environment, and influence daily living activities [28,29]. On the other hand, a positive relationship between physical activity (e.g., moderate aerobic exercise or coordinative activities) and cognitive performance in youth has been found [25]. An earlier study inclusively showed a small-to-medium positive effect of physical activity on children’s cognitive outcomes and academic performance [30]. Furthermore, a previous narrative review developed by Logan, Harris, Duncan, and Schofield [31], summarized evidence of the efficacy of HIIT in adolescent health. The authors reported meaningful evidence supporting HIIT as a potentially efficacious exercise modality for use in amoung adolescents. Nonetheless, it also recognized a need to explicitly report between-group differences for HIIT intervention and the control groups or steady-state exercise, such as the magnitude of difference between HIIT and other exercise modalities being of great interest to public health. An earlier narrative review presented by Tomporowski, Davis, Miller, and Naglieri [32] included studies of the effects of physical exercise on cognition and academic performance in children. The latter authors reported that physical exercise could be a fundamental strategy to improve mental functioning characteristics, which are essential to cognitive development. Nonetheless, the outcomes of the studies revealed variability, and a weak selected outcome measure was exposed. This may be due to the researchers selecting populations that are not representative of the general population [32].

Due to the interest in the potential for physical exercise in its numerous practices to develop cognitive performance, it was considered that a systematic and rigorous approach to review the literature was necessary, enabling a robust summary of the knowledge on this important thematic. This present review aimed to synthesize and analyze the effects of HIIT on cognitive performance and psychological outcomes in the healthy young population.

## 2. Materials and Methods

This systematic review was completed and reported in accordance with the Preferred Reporting Items for Systematic Reviews and Meta-Analysis (PRISMA) statement guidelines [33].

### 2.1. Search Strategy

A comprehensive search of all electronically archived literature published was conducted in four electronic databases, namely: ISI Web of Science, PubMed, Scopus, and PsycINFO. The search was performed using the Boolean search method, which limited the search results with operators including AND/OR to only those researches containing relevant key terms in the scope of this review. The main categories of search terms were identified: “psychological” OR “behavioral” OR “cognitive outcomes” OR “attention” OR “concentration” AND “young” OR “adolescent” OR “children” OR “childhood” OR “youth” AND “high-intensity interval training” OR “HIIT” OR “vigorous-intensity training” OR “physical exercise”. Relevant research articles published between January 1975 through February 2021 were collected. Appendix A Appendix A reports the search strategies used in the four databases.

### 2.2. Study Selection and Eligibility Criteria

The initial search identified 7138 articles with potential relevance. After the removal of duplicates and studies that did not apply a high-intensity interval training protocol, a manual screening according to the title and abstract was performed, and those that were not relevant were excluded, followed by a restoration of full texts for evaluation by two authors (A.R.A. and R.D.). The different phases of the systematic review were described using the PRISMA statement [34], where this maps the number of records identified, included, and excluded, and the reasons for exclusions. Studies were included or excluded using criteria defined with the PICO (Population, Intervention, Comparison, and Outcome) principles [35]. The literature searches incorporated as inclusion criteria: (i) studies with the healthy untrained young population; (ii) studies that verify the effects of HIIT in psychological outcomes (i.e., self-concept, self-efficacy, anxiety) and cognitive performance (i.e., selective attention, executive function, concentration); (iii) randomized clinical trials with accurately measures and a HIIT program design. The exclusion criteria were: (i) studies with adults or elderly population or athletes; (ii) sample with physical disabilities, chronic, neurologic, or clinical diagnosis of attention deficit hyperactivity disorders; (iii) studies with low or moderate training intensities programs and with no HIIT programs. Following these criteria, 342 original research articles were full-text assessed for eligibility. For the qualitative analysis, 8 articles were included. Theses, dissertations, and conference abstracts or proceedings were also excluded. There were no restrictions on written language, but studies were required to have an English abstract and be published in a peer-review journal. A detailed flow chart including systematic literature search, screening, eligibility, and inclusion is shown in Figure 1.

### 2.3. Data Extraction and Synthesis

From the included articles, information on sample size, age, country, HIIT strategies, measurements and the main results (effect sizes and improvements) and conclusions of psychological outcomes and cognitive performance were obtained. The data were extracted by two authors (A.R.A. and R.D.), and inconsistent data were resolved by the third author (H.P.N.). The degree of association was interpreted while using Cohen’s d [36].

### 2.4. Data Analysis

#### Assessment Risk of Bias

The risk of bias was assessed by the Cochrane Reviews methods [37]. Two authors, (A.R.A. and R.D.), independently assessed the risk of bias of each study against key criteria: random sequence generation, allocation concealment, blinding of outcome assessment, blinding participants and personnel, incomplete outcome data, selective reporting, and other bias. The following classifications were used: low risk, high risk, or unclear risk. The authors resolved disagreements by consensus, and a third author (H.P.N.) resolved their disagreements if necessary. Review Manager Software (RevMan, The Nordic Cochrane Centre, Copenhagen, Denmark) Version 5.4 was used to create the risk of bias graphs.

### 2.5. Statistical Analysis

The results of the included studies were recalculated to determine the effect sizes as a measure of the difference between averages in terms of standard deviation units, which offers evidence about the magnitude of the observed relationship between factors [38]. Accordingly, this analysis was estimated using Cohen’s d [36], where the mean experimental value was subtracted from the mean control value and divided by the combined standard deviation. This method permitted to determine the magnitude effects of differences between experimental conditions for the studies that provided means and standard deviations. The magnitude of the effect was classified as small (d = 0.2), intermediate (d = 0.5), or large (d = 0.8) [36].

## 3. Results

### 3.1. Description of the Studies Reviewed

A detailed analysis of these studies was reported (Table 1). The age of participants in all articles was under 18 years old and included boys and girls. The sample came from four different countries: 2 studies in Australia [20,39], 3 studies in Spain [21,22,40], 1 study in Japan [23], and 2 studies in the United Kingdom [24,41]. These studies included sample sizes from 30 to 184 subjects, ranging in age between 8 and 16 years old. For the study design, all the studies included a randomized controlled trial design. From the 8 studies reviewed, 62.5% (*n* = 5) developed a chronic intervention (i.e., repeated sessions of HIIT throughout days, weeks, or months), and 37.5% (*n* = 3) applied an acute intervention (i.e., single sessions of HIIT). Regarding the modality of HIIT, 87.5% (*n* = 7) of the studies [20,21,22,23,24,40,41] applied a traditional HIIT, in terms of running, sprinting, jumping, whereas 12.5% (*n* = 1) of the studies [39] used a traditional HIIT and also a high-intensity functional circuit training. Concerning the variables studied, 50.0% (*n* = 4) of the studies focused on cognitive performance [21,22,23,40], 37.5% (*n* = 3) of the studies targeted the psychological outcomes [20,24,41] and 12.5% (*n* = 1) of the studies determined the cognitive performance and psychological outcomes [39].

### 3.2. Risk of Bias in the Included Articles

About 50.0% of the studies were randomized and 50.0% used a crossover design. Most investigations did not implement a blinding design, and most of the studies made a between-group comparison. In fact, the blinding item is identified as the lesser item applied, due to inherent difficulty for practical reasons [42]. Only 25.0% of the studies revealed their concealed allocation, which would conduct itself toward systematic bias of therapeutic effectiveness [42]. About 75.0% of the studies reported a low risk of bias in the incomplete outcome data (attrition bias domain), which revealed transparency in the methodology used, and that well reported losses and exclusions occurred in the studies [37] (Figure 2 and Figure 3).

### 3.3. HIIT in Children and Adolescents Cognitive Performance

The cognitive performance of children and adolescents was observed in 4 studies [21,22,23,40], producing a total of 9 intervention effects (Table 1). One study found a positive effect of HIIT (4 weeks, 3 sessions/week, ≥85% HRmax, aerobic and core exercises) on executive function [23]. Another study [22] reported a positive small effect of HIIT (12 weeks, 2 sessions/week of HIIT, >85% HRmax, work-to-rest 20 s:40 s to 40 s:20 s) on linguistic reasoning, concentration, and selective attention. Mezcua-Hidalgo and colleagues [21] found positive effects through a large and medium effect size of HIIT (single session of HIIT, combination cardiorespiratory and coordinative exercises, work-to-rest 30 s:30 s) on selective attention and concentration, respectively. Finally, Ardoy et al. [40] showed significant effects on non-verbal, verbal and numerical abilities, as well as to abstract and verbal reasoning, and a medium effect size on spatial ability when physical education classes were combined with high intensity training (16 weeks, 4 sessions/week).

### 3.4. HIIT in Children and Adolescents Psychological Outcomes

The psychological outcomes on children and adolescents were studied in 3 studies [20,24,41], yielding a total of 2 intervention effects (Table 1). One of those 3 studies found a positive effect of HIIT interventions (12 weeks, 3 sessions/week, ≥85% HRmax) on self-concept [20]. Another study [41] reported remarkable results through a positive medium and small effect size of HIIT intervention (single session of HIIT ≥ 85% HRmax) in the psychological well-being of boys and girls, respectively. Malik and colleagues [24] observed positive effects of HIIT intervention (single session, at 85% peak power) on children’s psychological well-being.

### 3.5. HIIT in Cognitive Performance and Psychological Outcomes

Interestingly, children and adolescents’ cognitive performance and psychological outcomes were analyzed in one [39] of the 8 studies included in the present review, which obtained different intervention effects (executive function, psychological well-being, physical self-concept). In this sense, a positive small effect was observed through two HIIT interventions (8 weeks, 3 sessions/week, protocol a: gross motor cardiorespiratory exercises; protocol b: combination cardiorespiratory plus bodyweight resistance training exercises) in executive function. As for psychological well-being, a positive small effect from both HIIT protocols was also observed. In the physical self-concept, a positive medium effect was obtained from one of the HIIT protocols (i.e., combination cardiorespiratory and bodyweight resistance training exercises) (Table 1).

## 4. Discussion

The present review aimed to analyze the effects of HIIT on cognitive performance and psychological outcomes in the healthy young population. The included studies identified substantial improvements in cognitive performance, executive function, selective attention, planning skills, reading speed, working memory, self-concept and psychological well-being using different HIIT strategies. An enhancement of student’s behavior in a learning context through a HIIT intervention (i.e., vigorous-intensity) was also observed when compared to moderate-intensity training (e.g., physical education classes). All studies implemented HIIT interventions (using cardiorespiratory, coordinative, or core exercises) with a period between 4 to 16 weeks, time of the protocol between 8 to 30 min of vigorous-intensity (≥85%HRmax), and found positive intervention effects on cognitive performance and psychological outcomes in the young population. In this sense, Mezcua-Hidalgo et al. [21] and Martínez-Lopez et al. [22], through a HIIT protocol of 20 min of a combination of cardiorespiratory and coordinative exercise (intensity of 85% HRmax), reported substantial improvements in selective attention, concentration and linguistic reasoning in young people aged 12–16 years. Ardoy et al. [40], by combining physical education classes with high intensity activities (four sessions/week for 16 weeks), showed positive effects in non-verbal, verbal and numerical abilities, as well as in abstract and verbal reasoning of the children. Tottori et al. [23] reported that 10 min of HIIT aerobic and core exercises (intensity of 85% HRmax) would also provide benefits to children’s executive function. On the other hand, Mayr et al. [20], who used 30 min of vigorous-intensity (≥85% HRmax) for 12 weeks (3 sessions/week), observed a positive small effect in self-concept. Malik and colleagues [24,41] indicated beneficial effects of HIIT (single session, 20 min, ≥85% HRmax) on self-efficacy and psychological well-being in adolescents.

In accordance with the results obtained in this present review, there is a recent study [43] which reported that physical activity could increase well-being through psychosocial mechanisms, being able to improve self-concept, self-esteem, cognitive ability, mental health and self-perception in youth. Hereupon, Hillman and colleagues [44] after a physical activity program (70 min of moderate-to-vigorous intensity) of five sessions/week for nine months revealed a positive effect of physical activity on the cognitive performance and brain function of children. However, Tottori et al. [23] reported significant benefits on children’s executive function through a four week HIIT intervention program (vigorous intensity) of three sessions/week for ten minutes. In this sense, previous studies provide pertinent evidence, emphasizing that vigorous physical activity programs could improve cognitive function [4], attention and memory [45] in children and adolescents. In fact, Venckunas et al. [46], after a HIIT intervention with intermittent running, three sessions/week for seven weeks, showed beneficial effects on attention in the young population. In this context, Mayr et al. [20] who applied a similar protocol of HIIT (i.e., intermittent running, with ≥85% HRmax), but with three sessions/week for 12 weeks reported improvements in the self-concept of children and adolescents. Jeyanthi et al. [47] also reported that a simple intervention of HIIT (≥85% HRmax) could provide benefits in attention, which is in line with the results obtained from Costigan et al. [39], who implemented a HIIT intervention with cardiorespiratory exercise mostly, for eight weeks, three sessions/week, work between eight to ten minutes, and achieved a positive effect on children’s attention. However, the above-mentioned studies exceeded the expectations established by Vanhelst et al. [48], concluding that it would be necessary to reach a threshold of >12 min/day of high-intensity exercise to improve attention capacity in adolescents of 12–17 years old. This inconsistency of results may be explained due to the age range in different studies.

Regarding cognitive function, different strategies could be considered to improve this capacity, such as applying a HIIT intervention for six weeks with five sessions/week [49]; applying a HIIT program (i.e., the intensity of 90% HRmax) involving one session/week for 12 weeks [50]; or even implementing a HIIT protocol (e.g., aerobic and core exercise, (≥85% HRmax) for four weeks with three sessions/week working at 8–10 min [23]. However, to obtain higher effects in memory, it seems to be pertinent to apply a HIIT protocol ≥ 8 weeks long [51]. HIIT could be accepted as a time-efficient method (involving short period of time, 30 s exercises at >85% HRmax intensity combined with recovery breaks of 30 s, with no equipment required), being able to provide significant positive effects on healthy children and adolescents’ cognitive performance and psychological outcomes.

This systematic review presents some limitations that must be recognized: (i) the small number of articles included in the final review and the exclusion criteria applied could be contributing to a limitation of the included studies; (ii) the risk of bias of the included studies reported that the majority of studies did not include a blind design. Moreover, only one study reported their concealed allocation. However, the accomplishment of the item can either be dependent on whether it was implemented in practice during the study or on the researchers’ difficulties to clearly expose the experiment reports. Regarding the blind design, the blinding item is considered as the lesser item applied due to inherent difficulty for practical reasons [42]. However, is seems pertinent to report the review’s strongholds. All studies were accurately analyzed in a way such that the psychological and cognitive tests, as well as the intervention programs, were described in detail (i.e., the methodology used, the frequency, intensity, type and time); it was also possible to provide evidence of the magnitude effects of differences between experimental conditions.

## 5. Conclusions

The studies included in this current review showed that different approaches to HIIT produce relevant and positive acute effects in different academic performance variables, as well as in behavior for learning. Thereby, HIIT interventions may be considered as a useful tool to fight the inactivity pandemic and also improve the cognitive performance of youth, which should be considered by physical educators, educators, sport science professionals, and researchers in their future work. However, the need for longitudinal research with longer follow-up periods was also identified. Nevertheless, all studies have shown positive significant effects of HIIT on cognitive performance and psychological outcomes on a healthy young population. Nonetheless, it is still unclear which chronic effects are produced with a long-term HIIT intervention in children and adolescents, as well as the uncertainty about the feasibility of HIIT in school-based programs. Therefore, further longitudinal research with HIIT interventions should be conducted to determine the effects of HIIT in academic performance (e.g., behavior for learning and teamwork skills) in a school context, and expand the wisdom regarding HIIT effectiveness in healthy youth. At this moment, considering the worldwide pandemic which has increased the time spent at home, it seems to be pertinent to challenge young populations to be active at home by doing simple and safe HIIT exercises of a vigorous intensity (>85% HRmax intensity, i.e., characterized by not being able to say more than a few words without pausing for a breath) combined with recovery breaks (work-to-rest ratio, 30:30 s) for 20 min, two sessions/week (examples of HIIT exercises can be consulted in Mezcua-Hidalgo et al. [21]).

## Figures and Tables

**Figure 1 ijerph-18-05344-f001:**
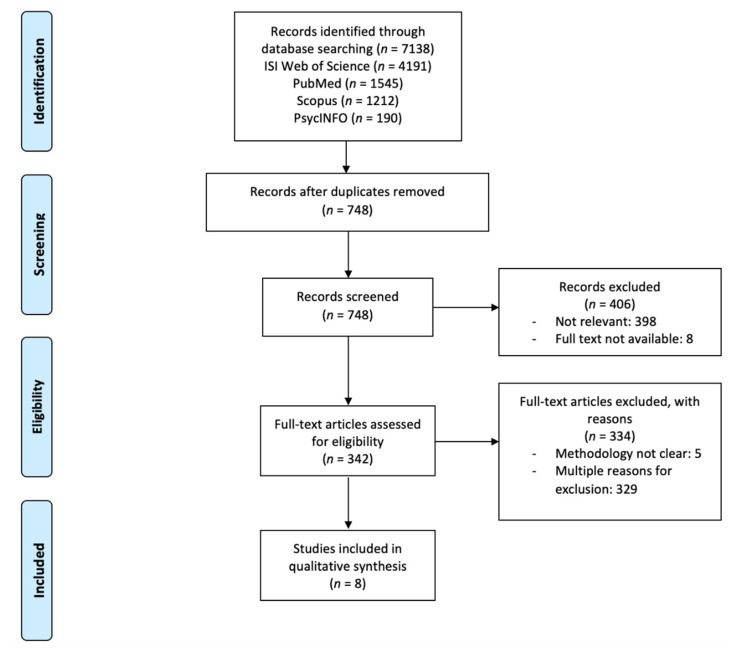
PRISMA flow chart of the search.

**Figure 2 ijerph-18-05344-f002:**
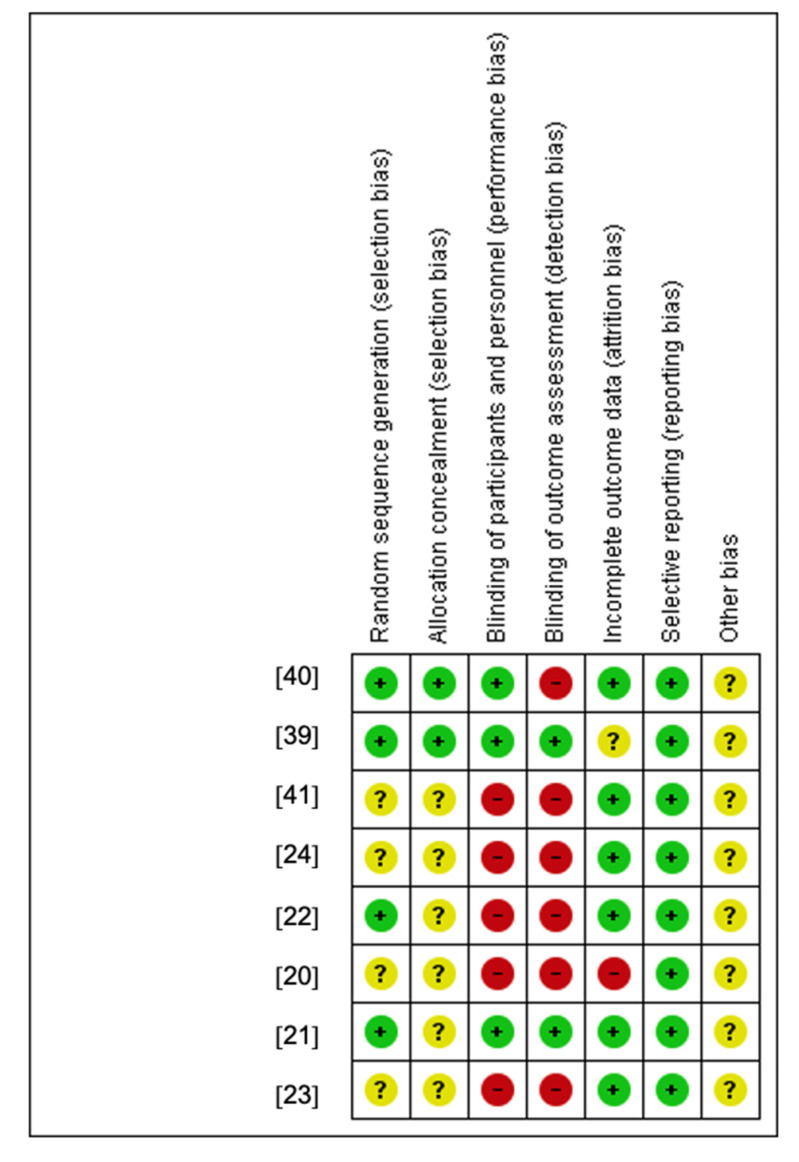
Judgments about each risk-of-bias item for each included study.

**Figure 3 ijerph-18-05344-f003:**
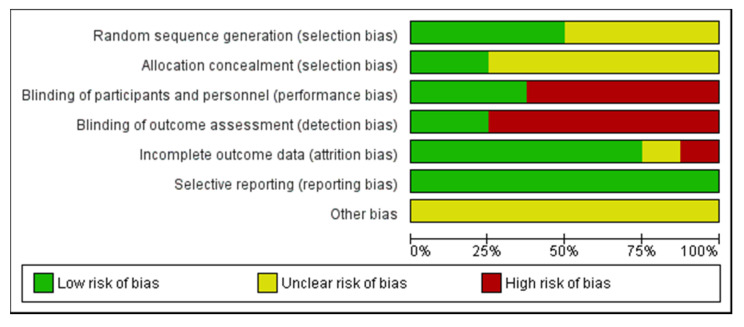
Risk-of-bias item presented as percentages across all included studies.

**Table 1 ijerph-18-05344-t001:** Characteristics of analyzed studies (*N* = 8).

Authors	Sample/Group/Age (Years)/Country	Exercise Protocol	Cognitive, Psychological and Behavior Measures	Main Outcomes
[39]	*N* = 65 EG1 = 21, EG2 = 22; C = 22 Age = 14–16 years Australian and New Zealand	Exercise Intervention EG1: WU + GMCardio + Stretch (work-to-rest 30 s:30 s) EG2: WU + CombCR + Stretch (work-to-rest 30 s:30 s) C: PE 3 sessions/week in 8 weeks	EF TMT—TA and TB PWB The Flourishing Scale PD Kessler Psychological Distress ScalePSC Physical Self-Description Questionnaire	EG1 EF (TMT B), d = 0.26, +11.45% EG2 EF (B-A), d = 0.28, + 17.73% | EF (TMT B), d = 0.39, +17.33% EG1 PWB, d = 0.19, +3.32% EG2 PWB, d = 0.21, +3.62% EG2 PSC, d = 0.50, +23.58%
[20]	*N* = 38 Age = 9–15 years Australia	EG1: intermittent fast running for shorts periods + long active recovery periods (30 min, HIIT, ≥85 HRmax) 3 sessions/week in 12 weeks	Psychological assessment Piers-Harris Children’s Self-Concept Scale	EG1 PSYA total score, d = 0.28, +8.71%
[24]	*N* = 30 Age = 11–13 years UK	EG1: 3′ WU at 20 W + 8 × 1-min work intervals at 85% peak power interspersed with 75 s active recovery at 20 W + 2′ Stretch at 20 W	Affective responses FS Perceived enjoyment During exercise EES; post-exercise PACES Perceived exertion Pictorial Children’s OMNI scale Behavioral activation and behavioral inhibition BIS and BAS	BAS/BIS with enjoyment responses PACES high BAS d = 0.55, +2.67% PACES low BIS d = 0.22, +1.35% Self-efficacy with enjoyment responses PACES high efficacy d = 0.55, +2.70% PACES low efficacy d = 0.83, +4.11%
[41]	*N* = 54 Age = 12–15 years UK	HIIT protocol: 3′ WU at 20 W + 8 × 1-min intervals at 90% peak power + 75 s recovery at 20 W + 2′ Stretch CMIE protocol: continuous moderate intensity cycling at 90% GAS)	Exercise Enjoyment Modified PACES for adolescents (perceived enjoyment)	HIIT protocol in PACES score (boys, d = 0.73, + 32.43%; girls, d = 0.39, + 11.58%)
[22]	*N* = 184 EG1 = 90, C = 94 Age = 12–15 years Spain	2 sessions/week in 12-weeks intervention EG1: 4′WU + 16′ over 85% HRmax within PE classes (work-to-rest between 20 s:40 s to 40 s:20 s) C: static stretch within PE classes)	Memory Ad hoc test of 1 min (RIAS test) Selective attention and concentration Brickenkamp’s d2 Test Linguistic reasoning Ad hoc test (reading speed and semantic comprehension)	EG1 selective attention, d = 0.29, +10.68% EG1 concentration, d = 0.28, +8.00% EG1 linguistic reasoning, d = 0.25, +7.66%
[21]	*N* = 158 EG1 = 77, C = 81 Age = 12–16 years Spain	Exercise Intervention EG1: 4′WU + 16′ combination cardiorespiratory and coordinative exercise (4 sets, 4 exercises, work-to-rest 30 s:30 s) C: static stretching	Cognitive Performance Ad hoc test 1 min (memory test) Brickenkamp’s d2 test (selective attention and concentration capacity) Measurements on baseline, immediately post, and after 2, 3, 4, 24, 48 h)	EG1 (after training) selective attention, d = 0.81, + 60.73% EG1 (after training, and 2 h after training) Concentration, d = 0.71, + 61.10%, and d = 0.72, + 62.49%
[23]	*N* = 56 EG1 = 27, C = 29 Age = 8–12 years Japan	Exercise Intervention EG1: 10′WU + 8′aerobic and core exercise + 5′Stretch (work-to-rest 30 s:30 s) C: PE 3 sessions/week in 4 weeks	Executive Function DFS/DBS test ToH	EG1 DFS test total score d = 0.33, +10.56% EG1 DFS test MS d = 0.22, +6.36% EG1 DBS test total score d = 0.30, +14.14% EG1 DBS test MS d = 0.34, +13.37% EG1 ToH 3-disk d = 0.75, +78.56% C ToH 4-disk d = 0.84, +66.18%
[40]	*N* = 67 EG1 = 26, EG2 = 23, C = 18 Age = 12–14 years Spain	EG1: PE, 4 sessions (55 min)/week for 16 weeks EG2: PE + high intensity training, 4 sessions (55 min)/week for 16 weeks C: PE, 2 sessions (55 min)/week for 16 weeks	Cognitive Performance IGF-M (non-verbal and verbal abilities, abstract reasoning, spatial ability, verbal reasoning and numerical ability)	Non-verbal abilities EG1, d = 0.39, +5.29% EG2, d = 0.88, +47.70% Verbal abilities EG2, d = 1.58, +22.61% Abstract Reasoning EG1, d = 0.34, +5.37 EG2, d = 0.75, +44.50% Spatial Ability EG2, d = 37.19, + 22.85% Verbal Reasoning EG2, d = 1.00, + 15.71% Numerical Ability EG2, d = 1.20, +8.28%

EG: experimental group; C = control group; PE: physical education classes; GMCardio: gross motor cardiorespiratory exercises; CombCR: combination of cardiorespiratory and body weight resistance training exercises; WU: warm-up; EF: execution function; PWB: psychological well-being; PSC: physical self-concept; PD: psychological distress; TMT: Trail Making Test; TA: Trail A; TB: Trail B; DFS/DBS test: Digit Span Forward/Backward test; ToH: Tower of Hanoi; GAS: gas exchange threshold; HIIT: high-intensity interval training; PACES: Physical Activity Enjoyment Scale; FS: feeling scale; EES: during exercise 7-point exercise enjoyment scale; PACES: post-exercise physical activity enjoyment scale; BAS: Behavioral Activation Scale; BIS: Behavioral Inhibition Scale; RPE: rating of perceived exertion; PSYA: psychological assessment; IGF-M: medium version of the Spanish Overall and Factorial Intelligence Test; d = effect size of Cohen’s d.

## Data Availability

This article was based on data from a Master Thesis in Physical Activity and Health.

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
