# Peer review of "High-Intensity Interval Training upon Cognitive and Psychological Outcomes in Youth: A Systematic Review"

_ijerph, 2021, doi:10.3390/ijerph18105344_

Round 1
Reviewer 1 Report
Thank you so much for editing according to the suggestions. The paper is ready to be published in its current version.
Author Response
Thank you very much for your positive words and work's recognition.
Reviewer 2 Report
Comments to "High-intensity interval training upon cognitive and psychological outcomes in youth: a systematic review":
- The abstract needs a general introduction to the research topic.
- What is the main contribution of this work? Who is it addressed to? Do you have to develop the previous literature to notice the gap?
- Put in keywords "systematic review"
- What are the starting hypotheses?
-Eliminate from fig 1 the box where you say that additional records N = 0. He supports with nothing
- In the third phase you say that for other reasons you exclude 329, so that only 8 documents are obtained. It is too small a number to carry out a review with guarantees from which adequate results can be obtained to advance this subject.
- In view of this, I suggest expanding the revision
Reviewer 3 Report
This review analyze the effects of high-intensity interval training on cognitive performance and psychological outcomes in the young populationand this area focus is interesting an the work is well organized
Despite the small number of articles included in the final analysis, I believe that the study selection process is very rigorous Therefore, my opinion on the review is positive
Author Response
Thank you very much for the time spent on reading and revising our manuscript, and also for your positive comments.Round 2
Reviewer 2 Report
Comments:
- Why do you say in line 74 that "A meta-analysis of 59 studies concluded ..." when in the title you say that it is a systematic review?
- The study is short if it does not widen the vision of the previous literature.
Author Response
Please see the attachment.

This manuscript is a resubmission of an earlier submission. The following is a list of the peer review reports and author responses from that submission.
Round 1
Reviewer 1 Report
The present article has significant and interesting content, thereby presenting great potential. However, there are some considerations that need to be addressed:
Introduction section:
It would be pertinent if the authors could explain in which way the purpose of the study and the respective results will be important to the scientific community.
Although the authors reported the importance of physical activity on providing positive effects on the neurocognitive outcomes (line 33) and the predominance of high physical inactivity in youth (line 39), more references could be added to reinforce these ideas.
Methods section:
Line 68:
It is suggested to change from: “This systematic review was completing and reporting in accordance with the Preferred…” to: …was completed and reported in accordance to the…
Line 74:
Change from: “Literature and comprehensive search of all electronically archived literature published was conducted…” to: Comprehensive search of all electronically archived literature published was conducted…
Results section:
The authors should develop the risk-of-bias of the included articles. They can follow the same structure that was used to report the blinding design item and to expose what happened with other items such as uncertain in allocation concealment.
Discussion section:
The authors can add the analysed information from the risk-of-bias as part of the study’s limitations.
Line 217:
It is suggested to change from: “It was also observed enhancement of students’ behavior in a learning context through a HIIT intervention (i.e., vigorous-intensity) when compared to …” to: An enhancement of students’ behavior in a learning context through a HIIT intervention (i.e., vigorous-intensity) was also observed when compared to moderate-intensity training (e.g., physical education classes).
Line 219:
Change from “All the studies implemented HIIT interventions (using cardiorespiratory, coordinative, or core exercises) with a period between 4 to 12 weeks, …” to: All studies implemented HIIT interventions (using cardiorespiratory, coordinative, or core exercises) with a period between 4 to 12 weeks, …
It would be interesting if the authors, based on the results of the present systematic review, could provide practical implications to adopt at home-based exercises for youth (especially given the current pandemic situation).
I would like to raise the question to the authors on why they did not perform the meta-analysis?
Author Response
Dear Reviewer,
We would like to thank you the time spent on reading and analyzing our manuscript. We thank you for your expert view and affirmative comment on our article. Your suggestions contributed to a relevant reflection on this research issue and upgraded our thoughts and knowledge about it. We made a detailed reading, all the notes were considered and the suggested modifications were performed.
Best regards.
Reviewer 1 (Rev1)
Introduction section:
Rev 1: It would be pertinent if the authors could explain in which way the purpose of the study and the respective results will be important to the scientific community.
Authors (A1): Thank you for your interest and comment. Few studies have reported the effects of HIIT in psychological variables in young population, hence contributing to the miscomprehension of the association between those variables in youth. Through the present review, it was possible to accurately analyze the psychological and cognitive tests as well the intervention programs described in detail and determine the effects’ magnitude of differences between factors. The present systematic review could be a major contributor to a better clarification of this matter within the scientific community.
Rev 1: Although the authors reported the importance of physical activity on providing positive effects on the neurocognitive outcomes (line 33) and the predominance of high physical inactivity in youth (line 39), more references could be added to reinforce these ideas.
A1: Thank you very much for your suggestion. More references were added in order to support those ideas, as suggested.
In the line 33 it was added the following reference:
de Greeff, J.W.; Bosker, R.J.; Oosterlaan, J.; Visscher, C.; Hartman, E. Effects of physical activity on executive functions, attention and academic performance in preadolescent children: a meta-analysis. J. Sci. Med. Sport 2018, 21, 501–507, doi:10.1016/j.jsams.2017.09.595.
To the line 39 new references were added:
Heradstveit, O.; Haugland, S.; Hysing, M.; Stormark, K.M.; Sivertsen, B.; Boe, T. Physical inactivity, non-participations in sports and socioeconomic status: a large population-based study among Norwegian adolescents. BMC Public Health 2020, 20, 1010, doi: 10.1186/s12889-020-09141-2
Stein, R.; Borjesson, M. Physical Inactivity in Brazil and Sweden – Different countries, Similar Problems. Arq. Bras. Cardiol.2019, 112, 2, 119-120, doi: 10.5935/abc.20190010
Methods section:
Rev 1: Line 68: It is suggested to change from: “This systematic review was completing and reporting in accordance with the Preferred…” to: …was completed and reported in accordance to the…
Line 74: Change from: “Literature and comprehensive search of all electronically archived literature published was conducted…” to: Comprehensive search of all electronically archived literature published was conducted…
A1: Thank you for your suggestions. Both sentences were changed accordingly. “This systematic review was completed and reported in accordance with the Preferred Reporting Items for Systematic Reviews and Meta-Analysis (PRISMA) statement guidelines”.
“Comprehensive search of all electronically archived literature published was conducted in three electronic databases, namely, Web of Science, PubMed, and Scopus.”
Results section:
Rev 1: The authors should develop the risk-of-bias of the included articles. They can follow the same structure that was used to report the blinding design item and to expose what happened with other items such as uncertain in allocation concealment.
A1: Thank you for your pertinent remark. The mention section was rewritten. “About 45% of the studies were randomized and 57% used a crossover design. Most investigations did not implement a blinding design, mostly of the studies made a between-group comparison. In fact, blinding item is identified as the less item applied due to inherent difficulty for practical reason (de Morton, 2009). Only 14.3% of the studies revealed their concealed allocation, which would conduct to systematic bias of therapeutic effectiveness (de Morton, 2009). About 71.4% of the studies reported low risk of bias in the incomplete outcome data (attrition bias domain), which were revealed transparency in the methodology used and well reporting losses and exclusions occurred in the studies (Higgins & Green, 2011).”
References used during revision:
de Morton, N.A. The PEDro scale is a valid measure of the methodological quality of clinical trials: a demographic study. Aust. J. Physiother. 2009, 55, 129–133, doi:10.1016/S0004-9514(09)70043-1.
Higgins, J.P.; Green, S. Cochrane handbook for systematic reviews of interventions, version 5.1.0; The Cochrane Collaboration, Ed.; 2011.
Discussion section:
Rev 1: The authors can add the analyzed information from the risk-of-bias as part of the study’s limitations.
A1: Thank you for your suggestion. New sentences were added to the limitations section:
(Added to line 245) “…must be recognized: i) limitations such as the small number of articles included in the final review and the exclusion criteria applied could be contributing to a limitation of the included studies; ii) the risk of bias of the included studies reported that the majority of studies did not include a blind design. Moreover, only one study reported their concealed allocation. However, the accomplishment of the item can either be dependent on whether it was implemented in practice during the study or on the researchers’ difficulties to clearly expose the experiment reports. Regarding the blind design, the blinding item is considered as the less item applied due to inherent difficulty for practical reasons (De Morton, et al., 2009).”
References used:
de Morton, N.A. The PEDro scale is a valid measure of the methodological quality of clinical trials: a demographic study. Aust. J. Physiother. 2009, 55, 129–133, doi:10.1016/S0004-9514(09)70043-1.
Rev 1: Line 217: It is suggested to change from: “It was also observed enhancement of students’ behavior in a learning context through a HIIT intervention (i.e., vigorous-intensity) when compared to …” to: An enhancement of students’ behavior in a learning context through a HIIT intervention (i.e., vigorous-intensity) was also observed when compared to moderate-intensity training (e.g., physical education classes).
Line 219: Change from “All the studies implemented HIIT interventions (using cardiorespiratory, coordinative, or core exercises) with a period between 4 to 12 weeks, …” to: All studies implemented HIIT interventions (using cardiorespiratory, coordinative, or core exercises) with a period between 4 to 12 weeks, …
A1: Thank you. The sentences were changed, as suggested.
“An enhancement of student’s behavior in a learning context through a HIIT intervention (i.e., vigorous-intensity) was also observed when compared to moderate-intensity training (e.g., physical education classes).”
“All the studies implemented HIIT interventions (using cardiorespiratory, coordinative, or core exercises) with period between 4 to 12 weeks, time of protocol between 8 to 30 minutes of vigorous-intensity (≥85%HRmax) and found positive interventions effects on cognitive performance and psychological outcomes in the young population.”
Rev 1: It would be interesting if the authors, based on the results of the present systematic review, could provide practical implications to adopt at home-based exercises for youth (especially given the current pandemic situation).
A1: Thank you for your proposal. Some sentences were added to the conclusion section in order to provide practical implications, as suggested.
“At this moment, considering the worldwide pandemic which increased the time spent at home, it seems to be pertinent to challenge young population to be active at home doing simple and safe HIIT exercises in a vigorous intensity (>85% HRmax intensity, i.e., characterized by not being able to say more than a few words without pausing for a breath) combined with recovery breaks (work-to-rest ratio, 30:30 seconds) for 20 minutes 2 sessions/week (examples of HIIT exercises can be consulted in Mezcua-Hidalgo et al., 2019).”
References used:
Mezcua-Hidalgo, A.; Ruiz-Ariza, A.; Suárez-Manzano, S.; Martínez-López, E.J. 48-Hour Effects of Monitored Cooperative High-Intensity Interval Training on Adolescent Cognitive Functioning. Percept. Mot. Skills 2019, 126, 202–222, doi:10.1177/0031512518825197.
Rev 1: I would like to raise the question to the authors on why they did not perform the meta-analysis?
A1: Thank you for your question and attentiveness. In fact, this systematic review did not conduct a meta-analysis due to the considerable heterogeneity across the included studies (e.g.: protocols, duration of the study, sample) (Ioannidis et al., 2008).
Reference used during revision:
Ioannidis, J.; Patsopoulos, N.; Rothstein, H. Reasons or excuses for avoiding meta-analysis in forest plots. BMJ 2008, 336,1413−1415.
Reviewer 2 Report
The present systematic review investigated the effects of high-intensity interval training (HIIT) on cognitive performance and psychological outcomes in the young population. Totally 7 articles were included in the review. The results suggested that HIIT would provide positive effects on cognitive performance and psychological outcomes in the young population. This is an interesting and important topic. Generally, the manuscript is well prepared. However, there are several concerns that should be addressed.
Major concerns:
- The rationale of the present study should be further highlighted. Although it was mentioned that related studies were limited in the youth population, it is still not clear why it is necessary to do this systematic review. Why do the authors want to investigate the cognitive performance and psychological outcomes? Why do the authors want to include children and adolescents only? Furthermore, the authors need to present the background in a more systematic way, e.g., highlight the effect of PA on cognitive performance and psychological outcomes in both children and adolescents. Also, a clear definition of “cognitive performance” and “psychological outcomes” is needed to clarify the outcomes of the present systematic review.
- Another concern of the present study was that the search strategy. Besides the database mentioned, the authors need to consider including some psychological databases. Also, using the current keywords may not be able to include all the related papers. Why not include papers published before 1998? Or the authors may need some references to support the usage of the current keywords.
- Considering the limited studies included (n=7), and both chronic and acute HIIT were summarized, the conclusion of the present systematic review is not so strong. The authors should consider performing a meta-analysis including subgroup analysis to convince the readers that the conclusion is strong and reliable.
- For the included studies, it seems that some studies did not include a control group or include PE class as a control group. Will this affect your discussion and conclusion?
- For the discussion, the authors may consider discussing the potential differences between HIIT and other types of PA programs, so as to highlight the strengths of adopting HIIT among this specific population, i.e., children and adolescents.
Author Response
Dear Reviewer,
We thank you for your valuable time and the constructive and helpful comments. We carefully addressed all of your concerns and suggestions in the following point-by-point statement. We hope that you will find our revision suitable to be accepted for publication in International Journal of Environmental Research and Public Health.
Best Regards.
Reviewer 2 (Rev2)
Major concerns:
Rev2: The rationale of the present study should be further highlighted. Although it was mentioned that related studies were limited in the youth population, it is still not clear why it is necessary to do this systematic review. Why do the authors want to investigate the cognitive performance and psychological outcomes? Why do the authors want to include children and adolescents only? Furthermore, the authors need to present the background in a more systematic way, e.g., highlight the effect of PA on cognitive performance and psychological outcomes in both children and adolescents. Also, a clear definition of “cognitive performance” and “psychological outcomes” is needed to clarify the outcomes of the present systematic review.
Authors (A1): Thank you very much for your pertinent remarks. Indeed only a few studies have reported the effects of HIIT in psychological variables in young population, hence contributing to the miscomprehension of the association between those variables in youth. Through the present review, it was possible to accurately analyze the psychological and cognitive tests as well the intervention programs described in detail and determine the effects’ magnitude of differences between factors. The present systematic review could be a major contributor to a better clarification of this matter within the scientific community.
Cognitive performance and psychological variables were included in the present review due to their influence on the daily living activities of the young population, the relationship between cognitive function and physical activity and also by the lack of knowledge regarding to the association between HIIT and psychological variables in youth. Given the last-mentioned reason, we justify why only children and adolescents were included in the review.
Some sentences were added to the introduction section of the manuscript, as suggested:
(added to the line 62) “Low level of cognitive performance during youth has been associated to psychological concern such as unkind emotions. Those emotions could produce depressive feelings, unhappiness or harmful interpretations of the environment and influence the daily living activities (Gale et al., 2008; Jaycox et al., 2009). On the other hand, a positive relationship between physical activity (e.g., moderate aerobic exercise or coordinative activities) and cognitive performance in youth has been found (Ruiz-Ariza et al., 2017). Cognitive performance could be described by several variables concerning to executive functions such as concentration, selective attention or working memory (Ruiz-Ariza et al., 2017). Psychological outcomes are obtained by variables describing to behavioral actions such as anxiety, depression, distress, well-being, and self-efficacy (Thomas et al., 2020; Brett et al., 2019).”
References used:
Gale, C. R.; Hatch, S. L.; Batty, G. D.; Deary, I. J. Intelligence in childhood and risk of psychological distress in adulthood: The 1958 National Child Development Survey and the 1970 British Cohort Study. Intelligence 2008, 37, 6, 592–599. doi:10.1016/ j.intell.2008.09.002
Jaycox, L. H.; Stein, B. D.; Paddock, S.; Miles, J. N.; Chandra, A.; Meredith, L. S.; Tanielian, T.; Hickey, S.; Burnam, M. A. Impact of teen depression on academic, social, and physical functioning. Pediatrics 2009, 124, 4, e596–e605. doi:10.1542/peds.2008-3348
Ruiz-Ariza, A.; Grao-Cruces, A.; de Loureiro, N. E. M.; Martínez-López, E. J. Influence of physical fitness on cognitive and academic performance in adolescents: A systematic review from 2005–2015. International Review of Sport and Exercise Psychology 2017, 10, 1, 108–133. doi:10.1080/1750984X.2016.1184699
Thomas, J.; Thirlaway, K.; Bowes, N.; Meyers, R. Effects of combining physical activity with psychotherapy on mental health and well-being: A systematic review. Journal of Affective Disorders 2020, 265, 475-485, doi: https://doi.org/10.1016/j.jad.2020.01.070
Brett, B.L.; Huber, D.L., Wild, S.B.A.; Noldon, L.D.; McCrea, M.A. Age of first exposure to American football and behavioral, cognitive, psychological, and physical outcomes in high school and collegiate football players. Sports Health 2019, 11, 4, 332-342. doi: 10.1177/1941738119849076
Rev2: Another concern of the present study was that the search strategy. Besides the database mentioned, the authors need to consider including some psychological databases. Also, using the current keywords may not be able to include all the related papers. Why not include papers published before 1998? Or the authors may need some references to support the usage of the current keywords.
A1: Thank you very much. The authors used different databases in order to include the majority of possible studies. PubMed is a database that focus mainly on medicine and biomedical sciences. Web of Science includes science, social sciences, arts and humanities, and Scopus physical sciences, health sciences, social sciences. The three databases cover most scientific fields. Using different databases guarantees the rigor of our analysis was applied and the non-exclusion of important studies to the systematic review.
Articles since 1998 were exposed because when the analysis was made the authors observed that the earliest study possible to be consulted was from the mentioned year. Concerning the keywords or terms applied, they were chosen after an exhaustive literature review, analyzing the terms of search that pertinent studies evolved in their research and also considering the major issues that the present systematic review want to analyze.
Rev2: Considering the limited studies included (n=7), and both chronic and acute HIIT were summarized, the conclusion of the present systematic review is not so strong. The authors should consider performing a meta-analysis including subgroup analysis to convince the readers that the conclusion is strong and reliable.
A1: Thank you for your remark. The current systematic review did not conduct a meta-analysis due to the considerable heterogeneity across the included studies (e.g.: protocols, duration of the study, sample) (Ioannidis et al., 2008). In order to discuss the origins of heterogeneity and to provide a detailed analysis in the present systematic review, the effects’ magnitude of differences between factors were determined. Moreover, new sentences were added to the conclusion section in order to strengthen it, as suggested.
(added to line 264) “At this moment, considering the worldwide pandemic which increased the time spent at home, it seems to be pertinent to challenge young population to be active at home doing simple and safe HIIT exercises in a vigorous intensity (>85% HRmax intensity, i.e., characterized by not being able to say more than a few words without pausing for a breath) combined with recovery breaks (work-to-rest ratio, 30:30 seconds) for 20 minutes 2 sessions/week (examples of HIIT exercises can be consulted in Mezcua-Hidalgo et al., 2019).”
References used:
Ioannidis, J.; Patsopoulos, N.; Rothstein, H. Reasons or excuses for avoiding meta-analysis in forest plots. BMJ 2008, 336,1413−1415.
Mezcua-Hidalgo, A.; Ruiz-Ariza, A.; Suárez-Manzano, S.; Martínez-López, E.J. 48-Hour Effects of Monitored Cooperative High-Intensity Interval Training on Adolescent Cognitive Functioning. Percept. Mot. Skills 2019, 126, 202–222, doi:10.1177/0031512518825197.
Rev2: For the included studies, it seems that some studies did not include a control group or include PE class as a control group. Will this affect your discussion and conclusion?
A1: Thank you for the pertinent question. Those mentioned facts were considered in the statistical analysis of the present systematic review. The results of the included studies were recalculated to obtain the effect sizes, thereby providing evidence about the magnitude of the observed relationship between factors (some examples: gender, group, time).
Rev2: For the discussion, the authors may consider discussing the potential differences between HIIT and other types of PA programs, so as to highlight the strengths of adopting HIIT among this specific population, i.e., children and adolescents.
A1: Thank you very much. Some sentences were added to the discussion section, as proposed.
(Added to line 219) Hereupon, Hillman and colleagues (2014) after a physical activity program (70 minutes of moderate-to-vigorous intensity) 5 sessions /week for 9 months revealed a positive effect of physical activity on the cognitive performance and brain function of children. However, Tottori et al., (2019) reported significant benefits on children’ executive function through a 4-week HIIT intervention program (vigorous intensity) of 3 sessions/week for 10 minutes.”
(Added into line 244) “HIIT could be accepted as a time-efficient method (involving short period of time, 30 sec exercises at >85% HRmax intensity combined with recovery breaks 30 sec, with no equipment require), being able to provide significant and positive effects on healthy children and adolescents’ cognitive performance and psychological outcomes.”
Reference used:
Hillman, C.H.; Pontifex, M.B.; Castelli, D.M.; khan, N.A.; Raine, L.B.; Scudder, M.R.; Drollette, E.S.; Moore, R.D.; Wu, C.; Kamijo, K. Effects of the FITKids randomized controlled trial on executive control and brain function. Pediatrics 2014, 134, 4, e1063-e1071, doi:10.1542/peds.2013-3219
Tottori, N.; Morita, N.; Ueta, K.; Fujita, S. Effects of High Intensity Interval Training on Executive Function in Children Aged 8–12 Years. Int. J. Environ. Res. Public Health 2019, 16, 4127, doi:10.3390/ijerph16214127.
Round 2
Reviewer 2 Report
Thanks for the efforts in responding to my comments and revising the manuscript. The quality of the manuscript has improved. There are several further comments for considerations.
Major concerns:
- The presentation on the rationale is still not very clear. For example, in page 2, line 54-56, the authors have mentioned that “less investigation has been reported”. The ref 21 is a systematic review published in 2015. I believe that more studies have been conducted on this topic during the past 5 years. Additionally, as the authors mentioned “the association between HIIT and psychological outcomes or behavior in youth remains unknown”, does it mean that there are no studies on this topic in the literature? If the answer is “yes”, it may not be appropriate to do this systematic review. If the answer is “no”, the authors should review the related studies. Therefore, a brief summary on the related studies will be helpful to highlight the research gap and/or rationale of the present study.
- For the search strategy, it is not ideal to include only these three databases although it seems to be acceptable. For the selected keywords, although the authors have provided the explanations, it is still questionable for the usage of these keywords. According to the Figure 1, the initial search only included around 200 papers covering such a broad topic. Therefore, references are necessary to support these keywords.
- The authors need to check all the references carefully. For example, Ref 30, 31 have been marked as included studies in Table 1. However, it is obvious they are not original studies and should not be included.